# *FMR1* and Autism, an Intriguing Connection Revisited

**DOI:** 10.3390/genes12081218

**Published:** 2021-08-06

**Authors:** William Fyke, Milen Velinov

**Affiliations:** 1SUNY Downstate Medical Center, SUNY Downstate College of Medicine, Brooklyn, NY 11203, USA; william.fyke@downstate.edu; 2Graduate Program in Neural and Behavioral Science, SUNY Downstate Medical Center, Brooklyn, NY 11203, USA; 3Rutgers Robert Wood Johnson Medical School, New Brunswick, NJ 08901, USA; 4Child Health Institute of New Jersey, New Brunswick, NJ 08901, USA

**Keywords:** fragile X syndrome, autism, neurodevelopmental disorders

## Abstract

Autism Spectrum Disorder (ASD) represents a distinct phenotype of behavioral dysfunction that includes deficiencies in communication and stereotypic behaviors. ASD affects about 2% of the US population. It is a highly heritable spectrum of conditions with substantial genetic heterogeneity. To date, mutations in over 100 genes have been reported in association with ASD phenotypes. Fragile X syndrome (FXS) is the most common single-gene disorder associated with ASD. The gene associated with FXS, *FMR1* is located on chromosome X. Accordingly, the condition has more severe manifestations in males. FXS results from the loss of function of *FMR1* due to the expansion of an unstable CGG repeat located in the 5′′ untranslated region of the gene. About 50% of the FXS males and 20% of the FXS females meet the Diagnostic Statistical Manual 5 (DSM-5) criteria for ASD. Among the individuals with ASD, about 3% test positive for FXS. FMRP, the protein product of *FMR1*, is a major gene regulator in the central nervous system. Multiple pathways regulated by FMRP are found to be dysfunctional in ASD patients who do not have FXS. Thus, FXS presents the opportunity to study cellular phenomena that may have wider applications in the management of ASD and to develop new strategies for ASD therapy.

## 1. Introduction

Neurodevelopmental disorders are a broad group of neurological and psychiatric conditions wherein the development of the central nervous system is disrupted. These disorders are typically recognized early in life and often persist into adulthood.

Of the many neurodevelopmental disorders, ASD has come to the forefront of social awareness. Recent estimates place the prevalence of Autism Spectrum Disorders (ASDs) at approximately 1 in 54 children, [1]. ASDs are characterized by persistent deficits in social communication and social interaction with restricted, repetitive patterns of behavior, interests, or activities [2]. These symptoms range from mild to severe impairments in daily functioning. It is important to note that severe impairments are more often seen in ASDs, as 78% of individuals diagnosed with ASD continue to require care into adulthood, whereas only 12% live independent lives [3]. Adding to the diversity of ASDs, a high rate of comorbidity with many other disorders appears to be the rule, rather than the exception [4]. Up to 79% of individuals with autism have motor delays, 12–70% depression, 45% intellectual disability (ID), 42–56% anxiety disorder, 9–70% gastrointestinal 0028GI) disturbances, 28–44% attention deficit hyperactivity disorder (ADHD), and 8–30% seizure disorder [5].

Twin studies have shown significant heritability of ASD [6,7], thus indicating a major contribution of genetic factors.

Nevertheless, a major obstacle in improving our understanding of the underlying pathology of ASDs has been the absence of identifiable biomarkers. With the development of advanced genomic technologies, many chromosomal copy number variants (CNV), as well as monogenetic mutations, were associated with ASD [8,9,10,11]. To date, hundreds of genes have been linked to ASD, however, most of these genes are involved in only a small percentage of cases [12,13]. A comprehensive genomic database that indexes and evaluates genetic ASD associations has been developed by the Simons Foundation Autism Research Initiative (SFARI) (https://gene.sfari.org/).

In addition to genetic links, exposures while in utero or during early development, such as toxins (e.g., lead, alcohol), medications (e.g., valproate, thalidomide), or stress, are strongly associated with neurodevelopmental disorders [14,15,16]. Risk-increasing genetic variants and early life exposures occur within unique genetic backgrounds and environmental contexts [17]. These factors combine to create an intricate etiological landscape, which presents unique challenges in comparison to other areas of medicine. This is particularly evident with regard to efforts to understand the pathophysiology and development of novel therapeutics.

A reliable biological definition of ASD would provide strong, distinct relationships between genomic variants, pathophysiological processes, and clinical phenotypes. However, many of the identified risk genes or environmental exposures are only found in a subset of patients and often share correlations with multiple disorders [18,19,20]. A similar situation is encountered with various pathophysiology at the cellular, circuit, and network levels [21].

A number of syndromic monogenetic neurodevelopmental disorders such as Fragile X Syndrome (FXS), Rett syndrome (RTT), and tuberous sclerosis (TSC), have strong associations with ASD [22,23]. On the surface, it appears logical to hypothesize that discrete genetic causes of neurodevelopmental disorders would provide relatively straightforward answers to questions about the underlying neuropathology. On the contrary, even in the case of monogenetic disorders, complex processes govern the downstream effects, as the loss of a single gene results in drastic effects on the activity of a multitude of other cellular processes [24,25].

Even though monogenetic neurodevelopmental disorders are distinct clinical entities, these disorders have provided important insights into potential common-shared pathophysiology and may provide a guide for unraveling the molecular underpinnings of non-syndromic autism [26].

One of the most important examples of syndromic ASD is Fragile X Syndrome. Approximately 30 to 50% of individuals diagnosed with FXS meet the criteria for a diagnosis of ASD, with FXS-ASD patients composing approximately 3% of all cases of ASD [27,28,29,30]. The co-occurrence of syndromic disorders, such as FXS, and the high level of heritability seen in twin studies, were among the first findings to make clear the importance of genes in the etiology of ASD [17,31]. FXS is the most common single-gene defect identified in patients with ASD. Accordingly, testing for the *fragile X mental retardation 1* (*FMR1)* gene mutation, the mutation responsible for FXS, is recommended as a first-tier genetic test in the current expert guidelines for ASD management [22].

## 2. Fragile X Syndrome

Fragile X syndrome (FXS) is a neurodevelopmental disorder due to an X-linked mutation in the *FMR1 gene*. The overall prevalence of FXS is approximately 1 in 7000 in males and 1 in 11,000 in females [32,33]. In more than 95% of known cases, the FXS phenotype is due to an expansion of more than 200 repeats and the subsequent methylation of CGG triplets in the 5′ untranslated promoter region of the *FMR1* gene [22]. The remaining 5% of FXS cases, in which triplet repeat expansions are not found, are often due to point mutations or deletions in the *FMR1* gene which, as in the other 95% of cases, result in absent or markedly decreased production of the fragile X mental retardation protein (FMRP). FMRP regulates the translation of approximately 4% of fetal brain mRNA and directly regulates several classes of ion channels [34,35,36,37]. Clinically, FXS patients present with distinct physical and behavioral features. Physically, individuals have characteristic facial abnormalities (e.g., elongated face, large ears), macroorchidism, hyperlaxity of joints, and hypotonia. Behaviorally, male FXS patients typically have intellectual disability (ID), anxiety, attention deficit hyperactivity disorder (ADHD), and sensory processing deficits, while female patients have more variable manifestations [22].

### 2.1. FMR1

Studies from the 1980s identified nonpenetrant male carriers in families with FXS, indicating a unique pattern of inheritance for the syndrome [38]. This peculiar pattern of inheritance remained unresolved until the *FMR1* gene was identified in 1991 by positional cloning [39]. The 5′-untranslated promoter region for the *FMR1* gene typically contains less than fifty-five CGG trinucleotide repeats and is located at Xq27.3 (Figure 1). Individuals with the full *FMR1* mutation have more than 200 repeats which result in methylation and subsequent gene silencing. Intermediate repeats of 55–200 are defined as premutation [22]. The presence or absence of FXS phenotypes from generation to generation is caused by anticipation, wherein an expansion of the premutation occurs during meiosis and produces the full mutation. Mothers with a premutation of greater than 90 to 100 repeats have up to 50% risk of passing on the full mutation to each of their children [40]. Males with full mutations are affected with FXS and typically present with intellectual disability, anxiety disorder, and attention deficit hyperactivity disorder, while full mutation females have variable and often milder manifestations (Hagerman et al., 2017). Due to the X-linked nature of this syndrome, females with the full mutation retain a functional copy of the *FMR1* gene and thus have milder phenotypes than their male counterparts. As a consequence, full mutation females are often diagnosed with FXS only after confirming the condition in a male relative, while some may never be identified [41]. Individuals with the premutation do not express the severe phenotypes seen with the full mutation however several disorders are associated with premutation carriers, namely, fragile X-associated tremor and ataxia syndrome (FXTAS), fragile X associated primary ovarian insufficiency (FXPOI), and fragile X premutation associated conditions (FXPAC)) [24].

### 2.2. FMRP

The product of the *FMR1* gene is the fragile X mental retardation protein (FMRP). This protein is highly conserved among vertebrates (~92% similarity between human and chicken homologs) and thus speaks to the importance of its biological activity [42]. FMRP is a 71 kDa protein containing a nuclear localization signal (NLS) domain, a nuclear export signal (NES) domain, three hnRNP-K-homology (KH) domains (KH0, KH1, KH2), and an arginine-glycine-glycine (RGG box) [43,44,45]. There are 12 identified isoforms of the FMRP protein that are produced as a result of alternative splicing [46]. The level of expression of each isoform varies during development and modulates binding affinities to ribosomes.

FMRP is diffusely expressed throughout the brain and varies widely among subpopulations of neurons [47]. The levels of expression of FMRP change dramatically during development, with FMRP expression reaching maximal levels early in the postnatal period, typically peaking in the early development and declining across early life [48,49]. Developmental studies with mice show a peak level of expression in the hippocampus, cerebellum, striatum, and brainstem, between postnatal days (P) 3 and P7 [50]. In the auditory and somatosensory cortex, peak FMRP levels are seen between P7 and P12. Unfortunately, differential studies of cell types and FMRP expression in the developmental brain have yet to be performed.

Within a given neuron, FMRP localizes to the soma, axon, and dendritic compartments [51]. FMRP assembles with RNAs, other binding proteins, and the homologous proteins FXR1P and FXR2P, to form large ribonucleoprotein complexes [52]. These complexes regulate the transport, translation, and metabolism of mRNAs [53]. This activity is crucial for appropriate neuronal development, synaptic connectivity and plasticity, and dendritic architecture [54,55,56,57].

Canonically, FMRP’s main function is that of an mRNA binding protein, responsible for the localization, stabilization, and translation of approximately 4% of fetal brain mRNA [37,39,58,59]. FMRP binds mRNAs through purine quartet motif regions and interacts with ribosomes via its KH domains [43,59,60,61]. Many of the mRNA transcripts that FMRP regulates are responsible for the production of synaptic proteins that are crucial for the high fidelity information processing at the synapse [57].

Non-canonical functions of FMRP, such as nuclear functions, microRNA interactions, and protein–protein interactions have been identified [34,35,36,37]. Of these functions, FMRP plays a critical role in modulating the behavior of several classes of neuronal voltage-gated ion channels [34,35,62]. Potassium channels such as Kv4.2, large conductance voltage and calcium-sensitive K^+^ (BKCa) channels, and Slack [34,63,64], calcium channels Ca_V_1.2, Ca_V_1.3 and Ca_V_2.2, [36,65,66,67], and non-selective hyperpolarization-activated and cyclic nucleotide-gated (HCN) channels [68,69] are modulated by FMRP activity. In the case of BKCa, Slack, Ca_V_1.2, and Ca_V_2.2, FMRP directly complexes with these channels to regulate their function.

## 3. *FMR1* and Autism Spectrum Disorder: Synaptic Dysfunction

There are many paradigms that can be used for developing research questions about the etiology of ASD. Currently, there is a vast array of relationships between ASD and disruptions in developmental processes due to genetic or environmental insults. Many of these perspectives provide useful frameworks for guiding research into potential causative mechanisms of ASD. One of the major areas of disruption in both FXS and ASD is found at the synapse [70,71,72,73,74] This has led to the proposal that ASD be conceptualized as a “synaptic disease” [73,75].

The loss of FMRP results in excess and dysregulated mRNA translation, delocalization of FMRP regulated proteins, and thus profound changes in the structure and physiology of the synapse [37,57]. A pivotal expansion in our understanding of the pathological effects of the *FMR1* mutation comes from the “mGluR theory” of FXS [76]. Type 1 metabotropic glutamate receptors (mGluR1 or 5) are G coupled protein receptors (GCPR) that are located post-synaptically and regulate multiple cellular signaling pathways [77]. Stimulation of mGluR5 receptors induces FMRP translation at the synapse and FMRP functions as a repressor of protein synthesis [51]. In the *Fmr1*-KO mouse, an mGluR5 regulated form of synaptic plasticity, long-term depression (LTD) is exaggerated [78]. The pathology seen in *Fmr1*-KO mice is reflected in both FXS and non-syndromic ASD patients, as alterations in mGluR5 expression are seen in postmortem ASD brains [79,80]. Additionally, high-throughput sequencing of mGluR signaling pathway genes has detected enrichment of rare variants among ASD patients [81]. Since dysfunctional mGluR activity is present in FXS and some ASD patients, this has facilitated detailed investigations into downstream components of mGluR5 signaling. Targeted mutations of mGluR5 scaffolding proteins such as *Homer1a, Shank3, Ngln3* produced phenotypes that approximate those seen in FXS and ASD [82,83,84]. Importantly, the mGluR hypothesis established a framework for investigations of synaptic dysfunction.

Many of the synaptic deficits which occur due to *FMR1* mutations are in mechanisms of plasticity, such as α-amino-3-hydroxy-5-methyl-4-isoxazole-propionic acid (AMPA) receptor expression, *N*-methyl-*D*-aspartate (NMDA) receptor localization, and endocannabinoid (eCB) signaling [82,85,86,87,88]. Notably, mutations in many of these components, particularly FMRP, mGluR, and NMDAR, disrupt critical regulatory mechanisms such as eCB regulation of presynaptic activity [82,85,87,89]. These deficits are also found in the *Fmr1*-KO mouse model and importantly, genetic variants for these proteins are found in non-syndromic ASD patients and are predictive of an ASD diagnosis [90,91,92,93].

Synaptic dysfunction can be subdivided into various subcategorizations (e.g., channelopathies) [64,94]. Each of these subcategorizations provides a framework for generating hypotheses and may be useful for identifying causal mechanisms and potential treatment targets. Here we focus on a subset of presynaptic regulatory mechanisms associated with *FMR1* and ASD. More specifically this review highlights the relationship between *FMR1* mutations and ASD with regard to the dysfunctional regulation of presynaptic activity by the endocannabinoid system (ECS), BKCa channels, and Ca_V_2.1 and Ca_V_2.2 channels.

## 4. The Presynaptic Hypothesis of FMR1 and ASD

Of the many pathophysiological processes associated with *FMR1* mutations and ASD, those which are imperative for appropriate presynaptic activity have a substantial body of clinical and preclinical evidence implicating them in the pathology of both disorders. At the synaptic level, presynaptic dysregulation results in aberrant neurotransmitter release and altered synaptic plasticity, which underlies the hyperexcitability seen at the circuit level [36,63,64,65,87]. Dysfunctional local circuits may underlie larger scale brain network dysfunction (e.g., connectopathy), and are often detected in ASD patients [95,96,97]. Here we review, several selected presynaptic regulatory components associated with *FMR1* mutations and ASD: the endocannabinoid system (ECS), and BKCa channels, and, P/Q and N-type Ca^2+^ channels [36,65,98,99,100].

In essence, the presynaptic hypothesis posits that presynaptic dysregulation causes computational errors at the synaptic level, resulting in circuit level and systems-level brain network dysfunction that manifests as neurodevelopmental pathology.

### Overview

The ECS, BKCa channels, and Ca_V_2.1 and Ca_V_2.2 channels are regulated by FMRP activity [35,36,65,85,101,102,103]. In the case of BKCa channels and Ca_V_2.1 and Ca_V_2.2 channels, FMRP interacts with these channels directly to inhibit calcium entry into the cell [35,104]. With regard to the ECS, FMRP controls the localization and translation of mRNA for DGL-α, the enzyme responsible for the production of the primary eCB, 2-AG [85]. These components make important contributions to the regulation of neurotransmitter release from the presynaptic neuron, each of which is linked to the FMRP activity, and along with FMRP itself, inhibits presynaptic Ca_V_2.1 and Ca_V_2.2 channels [36,65,98] (Figure 2A). When FMRP function is lost as a result of *FMR1* mutations, or when there is loss of function in the aforementioned FMRP associated synaptic components, dysregulation of calcium dynamics occurs resulting inappropriate neurotransmitter release (Figure 2B) [35,36,65,84,86,101,105,106,107].

The presynaptic hypothesis is a reductionist paradigm for guiding research directed at a small subset of pathogenetic associations. It is not exclusive of other paradigms, or dysfunction in other pathways not discussed here. ASD can, and does, arise from numerous developmental insults. Each of these components has notable pre-clinical and clinical evidence linking them to lost FMRP function and ASD [36,65,85,90,100,109,110,111]. Importantly, these components can be manipulated genetically and pharmacologically to induce or rescue some ASD phenotypes, which makes them attractive targets for inquiry into underlying pathology, and potential therapies [85,99,108,112,113].

## 5. ECS

The endocannabinoid system (ECS) is composed of two primary cannabinoid receptors, cannabinoid type 1 receptor (CB1) and cannabinoid type 2 receptor (CB2), and two primary ligands, arachidonoyglycerol (2-AG) and N-arachidonoylethanolamine (AEA) [114,115,116]. CB1, a G-coupled protein receptor (GCPR), is expressed extensively in the central nervous system, with higher levels of expression found in the hippocampus, amygdala, striatum, and cortex [117]. CB2, also a GCPR, is expressed at low levels in the CNS and largely on microglial cells where they mediate immune responses [115,118]. Endocannabinoids are hydrophobic lipids that are biosynthesized and released on demand, unlike the majority of neurotransmitters, which are water-soluble, synthesized in advance, and stored in vesicles [119].

Of the two endocannabinoid ligands, 2-AG is the most abundant found in the mammalian CNS and is a full agonist at CB1 [120,121,122]. 2-AG synthesis follows two distinct mechanisms: First (eCB_mGluR_), activation of group I mGluR which activates phospholipase C β (PLC-β) to cleave phosphatidylinositol-4,5-bisphosphate (PIP_2_) to produce the 2-AG precursor, 1,2-diacylglycerol (DAG). This is hydrolyzed by the serine lipase, diacylglycerol lipase α (DGL-α) in central neurons and diacylglycerol lipase β (DGL-β) in immune cells (e.g., microglia, macrophages), to form 2-AG [123,124]. The second mechanism for 2-AG synthesis is dependent on rapid increases of intracellular Ca^2+^ via NMDA receptors (eCB_NMDA_) [125,126]. PLC-β is activated in a Ca^2+^ dependent manner and produces the precursor, DAG, needed for the production of 2-AG by DGL-α [127]. The synthesis of 2-AG in post-synaptic neurons occurs within a supramolecular complex wherein mGluR5 receptors are bound to Homer1a scaffolding proteins which also bind PLC-β and DLG-α resulting in rapid and spatially localized 2-AG synthesis [85]. Approximately 85% of 2-AG is hydrolyzed into arachidonic acid (AA) and glycerol the presynaptic enzyme monoacylglycerol lipase (MAGL), with the remaining 15% metabolized by the enzymes α/β-hydrolase-6 (ABHD6) and α/β-hydrolase-12 (ABHD12) [128,129]. The second endocannabinoid, AEA, is a partial agonist at CB1 [130,131]. AEA synthesis occurs in a Ca^2+^ dependent manner, in response to an influx on intracellular Ca^2+^ causes cleavage of phosphatidylethanolamine (PE) by N-acetyl-transferase into the AEA precursor, N-arachidonoyl-PE (NAPE), which is then cleaved by the NAPE-hydrolyzing phospholipase D (NAPE-PLD) into AEA. Metabolism of AEA is carried out by fatty acid amide hydrolase (FAAH) which hydrolyzes AEA into AA and ethanolamine (EA). Once synthesized, 2-AG and AEA diffuse retrosynaptically and interact with CB1 receptors located on the presynaptic neuron [130].

CB1 signaling by 2-AG or AEA can result in the activation of multiple signaling pathways mediated by the G_i/o_ protein subunits of CB1. CB1 activation inhibits adenylyl cyclase and reduces cAMP production [132]. Activation by CB1 agonists also induces mitogen-activated protein kinase (MAPK) and PI3K/AKT pathways which control gene transcription and cellular activity [133]. Crucially, CB1 inhibition of neurotransmitter release, responsible for synaptic plasticity, is mediated by G_i/o_ protein inhibition of presynaptic Ca_V_2.1 and Ca_V_2.2 channels [98].

The retrograde nature of the ECS provides a unique form of synaptic plasticity called depolarization-induced suppression of inhibition (DSI) at inhibitory GABAergic synapse and depolarization-induced suppression of excitation (DSE) at excitatory synapses [134,135]. Briefly, depolarization at the post-synaptic neuron induces the production of eCBs which act retrosynaptically to inhibit neurotransmitter release from the presynaptic neuron.

Endocannabinoids also exhibit activity at transient receptor potential cation channel subfamily V member 1 (TRPV1), G protein-coupled receptor 18, 55, and 119 (GPR 18; GPR55; GPR119) [136,137]. While the activity of these ligand–receptor interactions is not yet fully understood, it has been shown that signaling at these receptors with exogenous cannabinoids may mediate some of the anxiolytic and anti-epileptic properties of these molecules [138,139]. The ECS also has a critical developmental role, as, during gestation, DGL-α mediated 2-AG-CB1 signaling is necessary for appropriate neurogenesis, neuronal migration, and axonal targeting [140,141].

### 5.1. ECS, FMR1, and ASD—Clinical Correlates

A growing body of clinical evidence associates the ECS with ASD phenotypes. Post-mortem studies of brain tissue from ASD patients indicated reduced expression of *CNR1*, the gene for CB1R [142]. Additionally higher expression of CB2R has been found to be upregulated in some children with ASDs [143]. Analysis of multiple genomic databases found variants in *CNR1* and *DAGLA*, the gene for DGL-α, were significantly associated with autism [144]. A series of studies investigated gaze to facial stimuli, a behavior frequently altered in FXS and ASD patients and found that polymorphisms in the *CNR1* gene modulate striatal responses and gaze duration to happy faces [145,146]. Two recent studies detected lower levels of circulating endocannabinoids in ASD patients relative to controls [147,148]. Risk-increasing variants for ASD, that are also associated with FXS, have been detected in synaptic proteins important for ECS function such as *GRM5, NGLN3, HOMER1A*, *SHANK3,* and several genes coding for NMDAR subunits [81,90,93,149,150,151,152,153,154]. Given the known role that mGluR5 dysfunction plays in FXS pathology and ECS activity, it is important to note that mutations in *GRM5*, the gene for mGlur5, are risk variants for ASD [81]. Furthermore, alteration in both mGluR5 and FMRP expression and signaling has been detected in non FXS ASD patients [79,80,155,156].

### 5.2. ECS, FMRP, and ASD—Preclinical Studies

Studies with the *Fmr1*-KO mice consistently show evidence of ECS dysfunction [85,87,101,102]. FMRP binds the mRNA of DGL-α and controls its appropriate translation and localization at the postsynaptic density (PSD) [85]. Loss of FMRP expression resulted in delocalization of DGL-α and dysfunctional 2-AG mediated plasticity. It was demonstrated that, in the absence of FMRP, mGluR5 stimulation fails to induce 2-AG production in the prefrontal cortex (PFC), and thus the mGluR hypothesis of FXS is tied to dysfunctional eCB activity. Of note, in utero exposure to valproate, a known risk factor for ASD, is linked to decreased levels of DGL-α mRNA [157].

Molecular and physiological studies indicated that appropriate eCB_mGluR_ production requires a scaffolding protein called Homer1a, which complexes mGluR5 and DGL-α [87,158]. In *Fmr1*-KO mice, interactions between mGluR5 and Homer1a are reduced and this is causal for hyperexcitability of cortical neurons and seizures [159]. Homer1a proteins also mediate mGluR5 and NMDA interactions, likely coordinating eCB_mGluR_ and eCB_NMDA_ forms of 2-AG synthesis [86,160]. These interactions are disrupted in *Fmr1*-KO mice and upregulation of Homer1a expression rescued cognitive deficits. Importantly, increasing the bioavailability of 2-AG normalized plasticity deficits and rescued the hyperactive, anxiety, and cognitive impairments phenotypes of the *Fmr1*-KO mouse [85,161]. Furthermore, enhancement of AEA availability rescued deficits in social approach, memory, and deficit frequently found in *Fmr1*-KO mice [112,162,163,164].

Pharmacological and genetic manipulations of specific ECS components strengthen the link between the ECS and ASD pathology. Mice with a targeted DGL-α deletion from direct pathway medium spiny neurons (dMSNs) of the striatum had impaired social interest and increased repetitive behaviors [165]. Mice with global DGL-α deletion showed increased anxiety, stress, and fear responses [166,167]. Additionally, optogenetic activation of basolateral amygdalar glutamatergic circuits, circuits that are inhibited by DGL-α mediated 2-AG production, caused social deficits in mice [168]. Importantly, pharmacological augmentation of 2-AG levels blocked these deficits in social behavior from occurring. Pharmacological inhibition of DGL-α activity at adulthood caused social impairments, a communication deficit, and repetitive behavior in C57BL6 mice [108].

Developmental studies show that a temporally orchestrated pattern of ECS expression and activity is imperative for appropriate brain connectivity [140,141,169,170,171]. The results of a postmortem study of brain tissue from various developmental times points revealed that CB1 and the enzymes responsible for endocannabinoid synthesis and metabolism (e.g., DGL-α, MAGL, FAAH) have distinct patterns of expression across development, particularly during neonatal and infancy age ranges [172]. This is further demonstrated by mouse studies wherein mice null for the CB1R have altered brain connectivity [173,174]. This appears to approximate a neurobiological phenotype frequently seen in patients with ASDs [97,175,176]. Genetic deletion of CB1 expression revealed deficits in social behavior, cognition, and repetitive behaviors [177,178,179]. Selective deletion of CB1 revealed that a loss of CB1 in glutamatergic, but not GABAergic, cortical neurons resulted in a reduction of social interest [180].

### 5.3. ECS and ASD—Potential Therapeutics

Clinically, phytocannabinoids (pCBs), plant-derived molecules with similar chemical structures as eCBs, have demonstrated success in the treatment of neurodevelopmental disorders. The pCB, cannabidiol (CBD) has FDA approval for the treatment of two forms of epilepsy: Dravet Syndrome and Lennox-Gastaut Syndrome [181,182]. In regard to FXS, a phase 1/2 study with CBD and FXS patients found that 12 weeks of treatment produced substantial reductions in hyperactivity, social avoidance, anxiety, and compulsive behavior [183]. Importantly the frequency of adverse events was low, and no serious adverse events were reported. Additionally, several prior case reports of FXS patients and CBD treatment reported improvement of symptoms [184]. Evidence indicates that CBD may be useful as a treatment for non-syndromic ASD [185,186,187,188]. Studies with the pCBs cannabidiol (CBD) showed an improvement in aggression, hyperactivity, sleep problems, speech impairment, seizures, and anxiety in ASD patients [185,189].

Cannabidivarin (CBDV), a propyl analog of CBD, has also shown promise as a treatment for ASDs [190]. A small study of 17 ASD patients and 17 matched non-ASD controls found that CBDV produced unique neurobiological effects in glutamate metabolism in the basal ganglia of ASD patients [191]. CBDV treatment in a mouse model of RTT rescued the social deficits that arise as a result of the *Mecp2* mutation [192]. Additionally, CBDV treatment in the valproic acid rodent model of ASD rescued ASD-like behaviors and restored ECS activity in the hippocampus [193]. Currently, a clinical trial, funded by the United States Department of Defense, is underway for CBDV treatment in ASD patients (Clinicaltrial.gov; NCT03202303).

Importantly, these molecules largely avoid the undesired psychotropic side effects that result from CB1 activation, strengthening their appeal as potential treatments for ASDs [194]. Studies indicate that these molecules act on the ECS, however, the mechanism of action for pCBs is not well understood and requires further studies [138,195,196,197,198,199].

## 6. Large Conductance Voltage and Ca^2+^ Sensitive K^+^ (BKCa) Channels

Large conductance voltage and calcium-sensitive potassium (BKCa) channels are expressed ubiquitously throughout the body, however, regulatory subunits of these channels are tissue-specific [200]. In the central nervous system, the β4 regulatory subunit is referred to as the neuronal auxiliary subunit and is the most abundant of the subunits expressed with BKCa channels in central neurons [201,202]. In the CNS, BKCa channels are expressed in most brain regions at presynaptic terminals, however higher levels of expression are found in the cortex, basal ganglia, hippocampus, and cerebellum [201,203].

Functionally, the α subunit of the BKCa channel opens in response to membrane depolarization and intracellular increases in Ca^2+^ [204]. It has a bimodal response to these events; opening to allow a large efflux of K^+^ ions (thus hyperpolarizing the membrane) and complexing with P/Q and N-type Ca^2+^ channels to inhibit Ca^2+^ entry and control neurotransmitter release [205,206]. Of these two stimuli, Ca^2+^ entry is the rate-limiting step for BKCa activation [207]. FMRP regulates the Ca^2+^ sensitivity of BKCa channels through direct interactions with the α and β4 subunits [35,208]. This reduces action potential duration, controlling neurotransmitter release and repetitive neuronal activity.

### 6.1. BKCa, FMR1, and ASD—Clinical Correlates

Genetic studies have uncovered a relationship between genetic variants for BKCa genes and ASD. Skafidas et al., 2014 [90] examined the occurrence of specific single nucleotide polymorphisms (SNPs) and a diagnosis of ASD. A genetic diagnostic classifier of 237 SNPs in 146 genes was used with 85.6% accuracy in predicting a diagnosis of ASD in a cohort of central European individuals gathered from two different databases: SFARI and Wellcome Trust 1958 Normal Birth Cohort (WTBC) databases. Two of the SNPs determined to be most effective at determining a classification of non-syndromic ASD vs. non-ASD were found in the *KCNMB4* gene, (β4 BKCa subunit), and *GRM5* gene, (mGluR5) were two of the three identified genes. This is particularly important in regard to the overlap between FXS and ASD since BKCa channel activity is directly regulated by FMRP at the β4 unit and mGluR5 dysfunction in FXS has been well established [35,76,99].

Two studies that investigated chromosomal abnormalities in ASD patients discovered a mild to moderate association between mutations in *KCNMA1* and a diagnosis of autism [209,210]. Additionally, mutations in the *KCNMA1* gene were identified in two patients with ASD and intellectual disability [100]. Genome analysis of the first patient discovered a balanced de novo translocation (9q23/10q22) resulting in haploinsufficiency for the α subunit, while the second patient revealed a single point mutation in the KCNMA1 gene which resulted in an ALA138VAL substitution.

BKCa dysfunction is also associated with other neurodevelopmental disorders. A patient with moderate to mild intellectual disability and febrile seizures was identified as having a mutation only in the β4 BKCa regulatory domain of FMRP [211]. Analysis of the family found a maternal and paternal history of learning problems, however, this specific mutation, being X-linked, was found only in the maternal genome.

### 6.2. BKCa, FMR1, and ASD—Preclinical Studies

Studies with the *Fmr1*-KO mouse demonstrated that loss of FMRP regulation of BKCa channels increased action potential duration [35,212]. Specifically, loss of FMRP increased the after-hyperpolarization phase (AHP) of the action potential, increasing neuronal excitability, presynaptic Ca^2+^ influx, and neurotransmitter release. Zhang et al., 2014 [64] showed that loss of FMRP was also responsible for downregulation of BKCa channel expression in the *Fmr1*-KO mice. Importantly, genetic upregulation of BKCa channels expression normalized the synaptic and circuit deficits in the *Fmr1*-KO mouse [212]. These factors were determined to be contributory for the sensorimotor hypersensitivity phenotype in the *Fmr1*-KO mouse, a frequently co-morbid feature of ASD. A genetic mouse model null for the BKCa α subunit gene (*Slo1)* was developed to explore the role of BKCa channels in neurodevelopmental disorders [213]. This study found that mice null for BKCa α expression had impaired sensorimotor and spatial memory, with normal locomotor activity. Currently, phenotyping of the social behaviors of the BKCa^−/−^ mouse has not been performed. Our recent pharmacological study using the BKCa channel inhibitor paxilline found that paxilline treatment induced unique social anxiety-type deficits during adulthood [113].

BKCa channels expressed outside the central nervous system respond to eCB signaling in vascular endothelial cells [214]. Additionally, in the trabecular meshwork of the eye stimulation of CB1 receptors was coupled to the activation of BKCa channels [215]. To the best of our knowledge, interactions between the ECS and CB1 in central neurons have not been investigated, and thus represent an area for future studies.

### 6.3. BKCa, FMR1, and ASD—Therapeutics

A BKCa channel agonist, BMS-204532 (BMS), was developed in 2002 for the treatment of ischemic stroke, however, it failed to demonstrate clinically significant therapeutic effects in phase III trials [216]. Since BMS has a favorable safety profile it is currently under investigation as a treatment of BKCa channelopathies. Detailed analyses of cells cultured from patients with ASD and BKCa mutations demonstrated that channel function could be rescued by BMS [100].

Studies with the *Fmr1*-KO mouse have demonstrated promise for BMS as a therapeutic for FXS. In an initial study, BMS treatment rescued social, cognitive, and anxiety phenotypes and normalized dendritic morphology in the *Fmr1*-KO mouse [99]. Two subsequent studies have demonstrated that BMS can rescue dendritic hyperexcitability and the increased self-grooming and sensorimotor hypersensitivity phenotypes of the *Fmr1*-KO mouse [64,217]. One of the challenges in using BMS clinically is the short half-life in brain tissue (t_1/2_ = 1.9) [216]. This would result in a difficult dosing schedule and therefore additional development is needed for molecule refinement. Despite these challenges, these preclinical studies strongly suggest that BMS or a next-generation BMS-derived molecule could provide a pharmacological intervention.

## 7. Ca_V_2.2

Ca_V_2.2 channels are key components of neurotransmission in the central nervous system and in the autonomic and sensory nervous system, and play a key role in early development [218,219]. These channels are expressed throughout the brain [220]. It is largely the α1 subunit that determines the kinetic and voltage-dependent properties of these channels [221]. FMRP interacts via its C-terminal domain to the linker region between the II and III domains of the α1 subunit [36,104]. This functions to control Ca^2+^ currents by reducing the expression of Ca_V_2.2 channels via proteasomal degradation. A follow-up study found that loss of FMRP resulted in increased Cav2.2 channel currents due to increased surface expression [65]. This contributed to neuronal hyperactivity.

### Ca_V_2.2 and ASD—Clinical Correlates

There are numerous links with sequence variants in voltage-gated calcium channel genes and ASD. Iossifov et al., 2014 [20] identified a de novo variant in the CACNA2D3 associated with ASD, a gene that acts as a regulatory subunit for Ca_V_2.2. A study of 20 ASD patients identified a duplication of the chromosomal region 9q43.3, which contains the gene CACNA1B, which produces Ca^2+^ currents in Ca_V_2.2 channels [222]. This duplication was found in 12 of the 20 patients. Altered Ca^2+^ activity itself has been identified in ASDs. A small study of six ASD patients with six age-matched controls found significantly higher levels of Ca^2+^ in ASD patients [223]. These findings established a link between ASD and Ca_V_2.2 channels and highlight their physiological importance in neuronal functions.

Due to the widespread expression of Ca_V_2.2 channels in the body, and the effects that calcium channel-directed medications have on the cardiovascular system, the development of therapeutics which seek to target these channels in central neurons face significant challenges. However, the connection to FMRP, ECS, and BKCa channels have with alterations in Ca_V_2.2 function and presynaptic activity represents an important pathway that may have therapeutic implications for ASD.

## 8. Clinical and Therapeutic Implications

In this article, we have reviewed a specific subset of deficits connected by *FMR1* mutations and ASD. Specifically, a set of neural mechanisms disrupted by the *FMR1* mutation that result in presynaptic dysregulation, and also share associations with ASD independently of *FMR1*. The link between these systems and neurodevelopmental disorders, specifically FXS and ASD, is a relatively recent area of research [190,224,225,226]. These mechanisms are amenable to both genetic and pharmacological manipulation, and thus, present an intriguing opportunity for elucidating a subset of causative mechanisms for ASD. Importantly, the ECS and BKCa channels show promising evidence that indicates potential as therapeutic targets for ASD.

Future studies are needed to explore more detailed mechanistic questions surrounding the presynaptic hypothesis of ASD with regard to the ECS and BKCa channels. Questions such as, Do the ECS and BKCa channels interact? It is well established that they each regulate the same presynaptic Ca^2+^ channels. It has not been explored if eCBs have activity at BKCa channels in central neurons, however, eCBs can modulate BKCa channels in cell culture [215]. Developmental studies addressing questions regarding FMRP expression, ECS, BKCa, and Ca_v_2.2 channel activity during critical periods in development are sorely needed. Additionally, it is unknown if specific genetic variants in these *FMR1* related regulatory mechanisms are able to produce a model of the spectrum of ASD phenotypes. In the case of *FMR1*, unique mutations that interfere with FMRPs regulatory function on BKCa are associated with unique neurodevelopmental phenotypes associated with ASD [100,211].

Due to the high level of heterogeneity seen in autism, it is imperative that systems that are mechanistically linked, and shown to modulate a spectrum of behavior, be thoroughly studied. This is particularly crucial in regard to ASD. Novel methods for modeling this disorder are needed, as are therapeutics. Therefore, the development of new models, and the identification of associated genetic variants in the population, is critical for improving our understanding of this complex and diverse disorder. The dysregulated presynaptic regulatory mechanisms discussed in this review present numerous targets which could be approached systematically to answer questions about presynaptic calcium dynamics and spectrum-like phenotypes that arise either due to *FMR1* mutations, or direct insults to regulatory systems.

## Figures and Tables

**Figure 1 genes-12-01218-f001:**
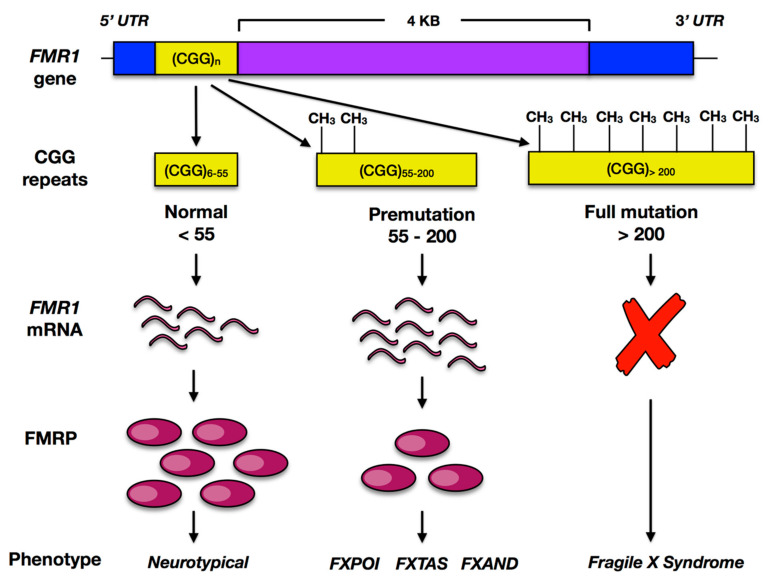
The FMR1 gene and Fragile X pathology. CGG repeats (yellow) in the promoter region. <55 repeats are typical. Repeat expansion resulting in the premutation (55–200) is found in 1/130–250 females and 1/260–800 males. The premutation expansion increases mRNA transcription and is associated with Fragile X primary ovarian insufficiency (FXPOI), Fragile X-associated tremor and ataxia syndrome (FXTAS), and Fragile X-associated neuropsychiatric disorder (FXAND). Repeats greater than 200 results in the methylation of the promoter region and gene silencing.

**Figure 2 genes-12-01218-f002:**
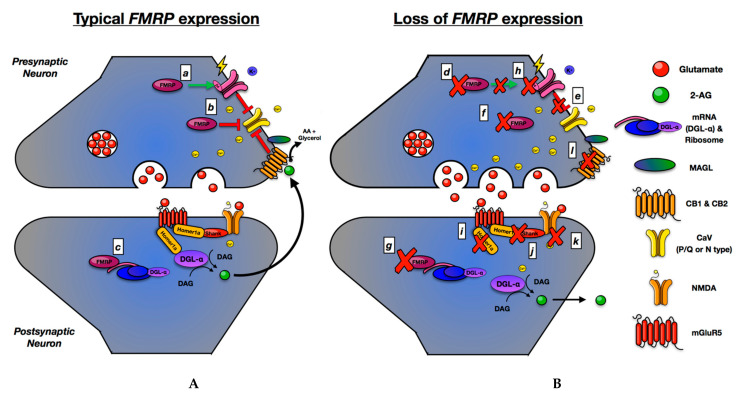
FMRP regulation of presynaptic activity. FMRP contributes to presynaptic regulation (**A**) by (a) stimulating the BKCa channel to inhibit CaV channels, (b) directly inhibiting Ca_V_ channels, or (c) controlling the translation and localization of the eCB producing enzyme *diacylglycerol lipase*
*α* (DGL-α) in the post-synaptic density. DGL-α exists in a complex (synaptosome) with the scaffolding protein Homer1a and mGluR5. DGL-α production of the CB1 ligand 2-AG occurs due to mGluR5 activity (Ca^2+^ independent) or NMDA activity (Ca^2+^ dependent). CB1 responds to 2-AG stimulation by inhibiting P/Q and N-type Ca_V_ channels. The absence of FMRP due to the *FMR1* mutation (**B**) results in a loss of appropriate presynaptic Ca_V_ channels regulation by (d,e) BKCa channels and (f) direct FMRP interactions; (g) absence of post-synaptic FMRP results in delocalized DGL-α and 2-AG production. Mutations in (h) the β4 regulatory unit of BKCa channels, (i) CB1, (j) Homer1a, (k) Shank3, and (l) NMDA channels have associations with syndromic and non-syndromic ASD. Each of these defects causes increased Ca^2+^ entry and neurotransmitter release (computational dysfunction). [108].

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
