# Peer review of "FMR1* and Autism, an Intriguing Connection Revisited"

_genes, 2021, doi:10.3390/genes12081218_

Round 1

Reviewer 1 Report

This is a well thought-out review that addresses and links two important clinical conditions (FXS and ASD). Below are some suggestions for improvement.

  1. In the last sentence of first paragraph the percentages for co-occurring conditions do not seem representative. Perhaps provide a range of presence of these conditions in ASD?
  2. Pg. 3, second paragraph. It seems the authors refer to females with the full mutation separately from females with FXS. How would one define females with FXS other than females who have the full mutation? Are the authors arguing that having the full mutation is not criteria for having a diagnosis of FXS? See Bartholomay, et al., Brain Sci. 2019, 9(1), 11; https://doi.org/10.3390/brainsci9010011
  3. Could Fig. 2 and Fig 3 be placed on the same page? This would facilitate comparison across the two and greatly increase readability. The key appears to be the same so some space could be saved if the two figures were merged and the key removed. 
  4. Table 1 is referenced on pg. 14 but I don’t see the table in the manuscript. 
  5. There are numerous grammatical errors throughout the manuscript. 

Author Response

  1. In the last sentence of first paragraph the percentages for co-occurring conditions do not seem representative. Perhaps provide a range of presence of these conditions in ASD?

    We are in agreement with the reviewer’s comments and have added ranges to the comorbid conditions associated with ASD. This now reads:

    “Up to 79% of individuals with autism have motor delays, 12-70% depression, 45% intellectual disability (ID)
    , 42-56% anxiety disorder, 9-70% gastrointestinal (GI) disturbances, 28-44% attention deficit hyperactivity disorder (ADHD), and 8-30% seizure disorder (Lai et al., 2014).”

  1. Pg. 3, second paragraph. It seems the authors refer to females with the full mutation separately from females with FXS. How would one define females with FXS other than females who have the full mutation? Are the authors arguing that having the full mutation is not criteria for having a diagnosis of FXS? See Bartholomay, et al., Brain Sci. 20199(1), 11; https://doi.org/10.3390/brainsci9010011

We are in agreement with the reviewer’s comments, that more definition needs to be given regarding the differences between males and females with the full mutation. We have added the following text and appropriate reference:

“Due to the X-linked nature of this syndrome, females with the full mutation retain a functional copy of the FMR1 gene, and thus have milder phenotypes than their male counterparts. As a consequence, full mutation female are often diagnosed with FXS only after confirming the condition in a male relative, while some may never be identified (Bartholomay et al., 2019).”.

  1. Could Fig. 2 and Fig 3 be placed on the same page? This would facilitate comparison across the two and greatly increase readability. The key appears to be the same so some space could be saved if the two figures were merged and the key removed. 

Fig. 2 and Fig. 3 have been merged into one figure (Fig. 2) and the legend has been revised in accordance with this.

  1. Table 1 is referenced on pg. 14 but I don’t see the table in the manuscript. 

    This has been removed.

  1. There are numerous grammatical errors throughout the manuscript. 

Grammatical errors have been identified and corrected.

Reviewer 2 Report

It is noted that some claims are provided in the abstract, which are not referenced in the body of the text, such as the 50% figure and the 3% figure. These should be addressed.

Page 1, a more recent summary than Baio is Maenner. CDC-MMWR 2020. Prevalence of Autism Spectrum Disorder Among Children Aged 8 Years — Autism and Developmental Disabilities Monitoring Network, 11 Sites, United States, 2016

This gives an overall prevalence of approximately 1 in 54.

 There must be a better estimate of the requirement for full time care than the 78% (Billstedt) ref as the CDC report indicates that about 2/3s of ASD subjects have an IQ of >70, many of whom can presumably live without full time care.

In my opinion, the Hallmayer and tick references have been largely supplanted with one by Bai. JAMA Psychiatry. doi:10.1001/jamapsychiatry.2019.1411.  Association of Genetic and Environmental Factors With Autism in a 5-Country Cohort

This larger survey gives an overall heritability estimate of around 80%.

Page 3.  FXAND and reference should probably be replaced with FXPAC as it is less pejorative regarding psychiatric disability. See Fragile X Premutation Associated Conditions (FXPAC). Johnson K, Herring J, Richstein J. Front Pediatr. 2020 May 27;8:266. doi: 10.3389/fped.2020.00266.

Author Response

Reviewer 2

  1. It is noted that some claims are provided in the abstract, which are not referenced in the body of the text, such as the 50% figure and the 3% figure. These should be addressed.

    Page 3 paragraph 5 now reads:

    “Approximately 30 to 50% of individuals diagnosed with FXS meet the criteria for a diagnosis of ASD with FXS-ASD patients composing approximately 3% of all cases of ASD
    (Kaufmann et al., 2017; Mendelsohn et al., 2008; Schaefer et al., 2008; Talisa et al., 2014).”

  2. Page 1, a more recent summary than Baio is Maenner. CDC-MMWR 2020. Prevalence of Autism Spectrum Disorder Among Children Aged 8 Years — Autism and Developmental Disabilities Monitoring Network, 11 Sites, United States, 2016

We have updated this part of the text with the recommended reference to read:

Of the many neurodevelopmental disorders, ASD has come to the forefront of social awareness. Recent estimates place the prevalence of Autism Spectrum Disorders (ASDs) at approximately 1 in 54 children (Maenner et al., 2020).”

  1. This gives an overall prevalence of approximately 1 in 54.

    Addressed above.

  2. There must be a better estimate of the requirement for full time care than the 78% (Billstedt) ref as the CDC report indicates that about 2/3s of ASD subjects have an IQ of >70, many of whom can presumably live without full time care.

    We have revised the text to read “78% of individuals diagnosed with ASD continue require care into adulthood, with only 12% live independent lives (Anderson et al., 2014; Billstedt et al., 2005)”. Additionally have added a second reference to a more recent study (Anderson et al 2014) which further supports the data from the study Billstedt et al 2005.

  3. In my opinion, the Hallmayer and tick references have been largely supplanted with one by Bai. JAMA Psychiatry. doi:10.1001/jamapsychiatry.2019.1411.  Association of Genetic and Environmental Factors With Autism in a 5-Country Cohort

Hallmayer et al citations have been replaced with Bai et al

  1. This larger survey gives an overall heritability estimate of around 80%.

Addressed above.

  1. Page 3.  FXAND and reference should probably be replaced with FXPAC as it is less pejorative regarding psychiatric disability. See Fragile X Premutation Associated Conditions (FXPAC). Johnson K, Herring J, Richstein J. Front Pediatr. 2020 May 27;8:266. doi: 10.3389/fped.2020.00266.

We are thankful for the reviewer’s comments and agree wholeheartedly. We have replaced FXAND with the more appropriate FXPAC,  and added the Johnson et al 2020 reference.